# Experimental Validation of a Permanent Magnets Magnetorheological Device under a Standardized Worldwide Harmonized Light-Duty Test Cycle

**Claudia Simonelli** [1,*] , **Luca Sani** [1] , **Nicolò Gori** [1] , **Miguel Fernández-Muñoz** [2] **and Antonino Musolino** [1] **and Rocco Rizzo** [1,*]

1    DESTEC Department, School of Engineering, University of Pisa, 56122 Pisa, Italy; luca.sani@unipi.it (L.S.); nicolo.gori@phd.unipi.it (N.G.); antonino.musolino@unipi.it (A.M.)
2    Signal Theory and Communications Department, Mechanical Engineering Area, Universidad de Alcalá, 28801 Alcalá de Henares, Spain; miguel.fm@uah.es
*    Correspondence: claudia.simonelli@phd.unipi.it (C.S.); rocco.rizzo@unipi.it (R.R.); Tel.: +39-0502217101 (C.S.); +39-0502217302 (R.R.)

**Abstract:** In this paper, the experimental validation of an innovative clutch based on magnetorheological fluids (MRFs) excited by permanent magnets is described. The device, used in automotive applications to engage and disengage the vacuum pump, is tested using a standardized Worldwide harmonized Light-duty Test Cycle (WLTC). A test bench is built, and the system is observed in its operation for one hour, considering two consecutive WLTCs. The temperature increase slightly impacts the clutch's behavior; in particular, the on-state performance of the device, mainly determined by the magnetic field-induced torque, remains largely unaffected by the temperature increase. The results showed that the performance of the proposed MRF-based device is only marginally affected by the phenomena that take place during the actual operation (e.g., temperature increase, shaft slip), confirming the effectiveness of the design.

**Keywords:** magnetorheological fluids; permanent magnets; MRF clutch; Worldwide harmonized Light-duty Test Cycle; vacuum pump





## 1. Introduction

In recent years, the automotive industry has seen a significant effort to enhance vehicle performance and efficiency, reducing consumption and emissions. In particular, a research field is focused on solutions to reduce the incidence of auxiliary device absorption (e.g., heating and cooling systems, hydraulic, oil, and vacuum pumps). An auxiliaries consumption reduction offers significant energy savings, especially dealing with large vehicles for public transportation.

The vacuum pump is used in almost all diesel and petrol engines to draw air from the auxiliaries that require working pressures lower than atmospheric (i.e., the power-brake unit, the air conditioning system, or the turbo system). A device connected to the vacuum pump is the power-brake system, placed between the brake pump and the pedal. It amplifies the force applied to the pedal by the driver, exploiting the difference in pressure between two chambers separated by a membrane. In normal driving conditions, both chambers are connected to the vacuum pump, and the pressure is below the atmospheric one (about 0.2 bar in steady state, in less than 20 s). When the driver exerts a force on the brake pedal, one of the chambers is connected to the atmosphere, producing different pressures on the two sides of the membrane. Usually, the vacuum pump is composed of a case within which two skids, led by a palette, can slide, lubricated by the engine oil. It is connected to the camshaft of the vehicle engine using a Holdam joint, and it operates continuously even if it is not needed.

In order to reduce this waste of energy, an innovative clutch based on magnetorheological fluids (MRFs) and permanent magnets (PMs) has been proposed in [1–7] to disconnect the vacuum pump from the camshaft when the desired pressure is reached in the power-brake chambers. Then, the magnetorheological fluid-based clutch (MRF-based clutch) reconnects the vacuum pump to the camshaft when needed again. A system of permanent magnets has been used to suitably excite the MRF, allowing the device to satisfy some constraints in the system specifications (e.g., volume limits, ON/OFF torque ratio, fail-safe operation). In these papers, authors proposed several devices with various dimensions and designs, considering different magnetizations (radial and diametral) of the magnets, gap arrangements, and ferromagnetic structures. In particular, the authors have analyzed the device using network approaches equivalent to those described in [8–10] and also used in [11–13].

The temperature dependence of the properties of MRFs has already been assessed in the literature [14–16]. In particular, the experimental characterization of the fluid behavior was performed at different temperatures, considering both excited and non-excited conditions. The correlation between the magnetic and mechanical properties and temperature has also been investigated in MR clutches, as thermal effects on torque proved to be significant [17,18]. These studies highlight a decrease in the transmissible torque when the temperature increases (up to 20% with an increase of 60 °C). Although experimental tests were performed on these devices, in most prior investigations, they were mainly observed during a single engagement/disengagement cycle to verify their capability.

In this paper, the experimental validation of the developed MRF-based clutch is described. The device was tested using a standardized Worldwide harmonized Light-duty Test Cycle (WLTC) performed on a test bench. The system was observed for one hour during its operation, considering two consecutive WLTCs. The results were analyzed to verify the effects of a standardized procedure on the device's performance.

Rheological fluids belong to a particular class of (smart) materials able to change their rheological behavior as a function of an external stimulus. In particular, in the electromagnetic area, it is possible to identify two main categories of smart fluids whose viscosity increases when an external field is applied: electrorheological fluids (ERFs), which respond to an electric field, and magnetorheological fluids (MRFs), firstly introduced by Jacob Rabinow, which react to a magnetic field [19].

These fluids are generally composed of micron-sized polarizable particles dispersed in a synthetic liquid medium; they exhibit a controllable transition from a liquid to a near-solid state upon the application of an external magnetic field. Typically, this transition occurs with the development of a yield/shear stress that monotonically increases with the amplitude of the external stimulus. Since the phenomenon is reversible, in a few milliseconds, the fluid returns to its original liquid state by removing the field [20–22].

The behavior of rheological fluids is often represented with a Bingham plastic model governed by the following equation:

$$\tau = \tau_0(H) + \mu\dot{\gamma}, \tag{1}$$

where $\tau_0(H)$ is the yield stress induced by magnetic field $H$, $\mu$ is the viscosity, and $\dot{\gamma}$ is the fluid share rate.

Typical MRF-based devices are used to absorb mechanical shocks and vibrations (e.g., dampers) or in haptics for the development of actuators with variable compliance [23,24] or to build interfaces capable of simulating objects in virtual environments [25–27].

Considering values of the yield stress, excitation sources, and response time, MRFs appear to be suitable for many applications revised in [28] together with the progress made in the recent past in fluid synthesis.

In the automotive or aerospace industry, several MR brakes or clutches have been developed. These devices allow a smooth and steady transmissible/braking torque, with many advantages over standard devices. For example, in [29], the use of an MRF-based brake fed with individual currents opportunely controlled allows power consumption

reduction, while the authors of [30] proposed an actuator integrated with the MRF able to act as a motor, clutch, and brake. The integration of magnetorheological fluids and actuators is also exploited in [31], where Xu et al. discussed the design and optimization of an axial flux PM device able to act as an actuator or brake. Moreover, the use of MRF-based brakes allows an increase in the device compactness, for example, exploiting the braking effects of the MRF produced with both shear and compression modes [32], but also in the performances of the vehicle they brake: the authors of [33] presented an MRF-based brake with a T-shaped disk excited by coils to improve motorcycle performance while braking. Another interesting MRF-based device is presented in [34], where the authors discuss the design and control of a clutch in which the fluid is excited both by PMs and coils. Even though the transmissible torque can be easily controlled by modulating the current flowing in the coils, demagnetization of PMs can occur, affecting the device's performance.

The novelty of the clutch presented in this paper is the use of a purely PM-based excitation system. This device is used to engage/disengage the vacuum pump of the braking system in automotive applications. The chosen structure, described in detail in the following section, fulfills the safety requirements, making the device fail-safe. Moreover, since the MRF viscosity is low in the OFF-state, low power losses are ensured in the disengaged condition. Furthermore, the MRF-clutch produces high transmissible torque in the engaged condition due to the high yield stress of the fluid when excited.

## 2. The MRF-Based Multi-Gap Clutch

The clutch described in this paper is the multi-gap MRF-based device shown in Figure 1 that comprises two shafts (the primary and secondary shaft, respectively), a double cup-shaped gap filled with MRF, and an excitation system consisting of PMs that moves axially and generates the magnetic field that excites the MRF. Two seals, positioned at low diameters, confine the fluid in the double cup-shaped area.

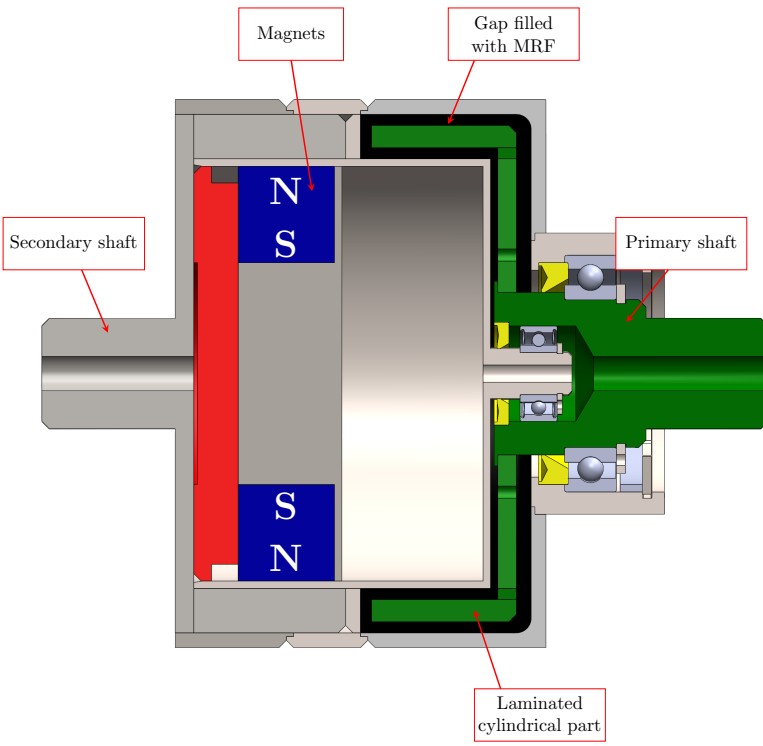

**Figure 1.** Arrangement of the multi-gap MRF-based clutch's components.

In this device, a commercial MRF, labeled MRF-140CG and produced by Lord Corporation© (Cary, NC, USA) [35], was used. It has a viscosity $\eta = 0.28$ Pa·s at 40 °C and a density between 3.54 and 3.74 g/cm$^3$. Yield stress can vary between $\tau_0 = 25$ Pa and

$\tau_{max} = 65$ kPa, when a magnetic field is applied as shown in Figure 2. When high fields are applied, the relative permeability becomes around 1.3, which implies a high magnetic reluctance. Its main magnetic and rheological features are derived from the datasheet and summarized in Table 1. Figure 2 shows the magnetic flux density and shear stress of the fluid as a function of the applied magnetic field.

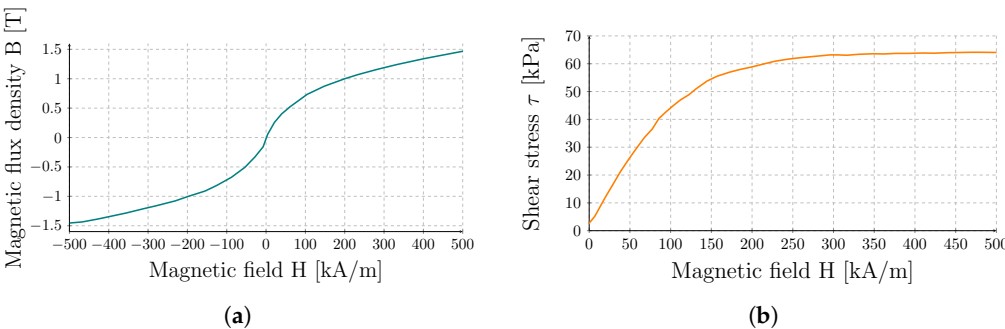

**(a)**　　　　　　　　　　　　　　　　　**(b)**

**Figure 2.** Magnetic (**a**) and mechanical (**b**) properties of MRF-140CG.

**Table 1.** MRF-140CG magnetic and rheological characteristics.

| Magnetic Properties | | Mechanical and Rheological Properties | |
|---|---|---|---|
| Saturation threshold $B_s$ | 0.6 T | Density $\rho$ | 3.54–3.74 g/cm$^3$ |
| Maximum excitation field $H_{max}$ | ~200 kA/m | Viscosity $\eta$ @ 40 °C | 0.280 Pa·s |
| Relative initial permeability $\mu_{r_0}$ | 5.5 | Initial yield stress $\tau_0$ | 25 Pa |
| Permanent relative permeability $\mu_{r_{perm}}$ | 1.3 | Maximum yield stress $\tau_{max}$ | 60 kPa |
| | | Solids content by weight | 85.44% |
| | | Operating temperature | $-40$ °C $< T < +130$ °C |

Figure 3 shows the excitation system composed of a NdFeB 90°-poles hollow cylinder magnetized along the diametral direction embedded in a chamber inside the secondary shaft. This configuration represents a good tradeoff between the increase of the transmitted torque with frequency (proportional to the number of poles of the PM-system) and the magnitude of the magnetic field inside the gap filled with the MRF [3]. Table 2 summarizes the main features of the PMs. The excitation system moves along the axial direction with an external pressure source that injects air through a hole in the primary and secondary shafts, which connects each part of the magnet chamber. In the case of pneumatic system failure, the engagement of the clutch is guaranteed with a preloaded spring that forces the magnets toward the fluid gap zone. This configuration assures fluid magnetization and, consequently, a fail-safe clutch operation.

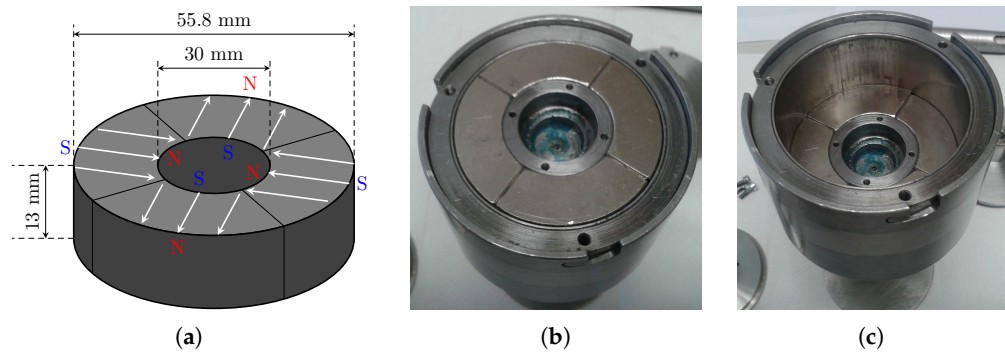

**(a)**　　　　　　　　　　　**(b)**　　　　　　　　　　　**(c)**

**Figure 3.** Excitation system that produces the magnetic field enclosed in the secondary shaft. Schematic view (**a**) and prototype in OFF-state (**b**) and ON-state (**c**) conditions.

**Table 2.** Permanent magnets features.

| Permanent Magnet Features | |
| --- | --- |
| Material | NdFeB |
| $B_r$ | 1.36 T |
| $H_c$ | $10.2 \times 10^5$ A/m |
| $T_{max}$ | 150 °C |

From a magnetic point of view, the arrangement of all ferromagnetic and non-ferromagnetic structures around the fluid was carefully designed to concentrate and confine the magnetic field within the region containing the MRF, maximizing the transmitted torque. The cylindrical part of the primary shaft consists of several isolated ferromagnetic layers of about 0.5 mm thickness made of AISI-1018 stacked together using non-ferromagnetic (AISI-304) mechanical joinings. Along the circumference of each layer, there are 36 internal and external semi-circular slots with a radius of 1 mm, created at the inner and outer diameters, respectively. The presence of the double squirrel cage allows a suitable address of the magnetic flux lines through both the cylindrical surfaces of the double cup-shaped gap filled with MRF. Figure 4 shows the geometry of a single ferromagnetic layer. Once the laminations are stacked, the slots are filled with a conductive alloy ($\sigma \simeq 4 \times 10^7 \ \Omega^{-1}/\text{m}$) and short-circuited in correspondence with the extremities in the axial direction using two rings of the same material. The secondary shaft is made of solid ferromagnetic material (AISI-1018), whose radial thickness is determined so that the magnetic induction is below the saturation level.

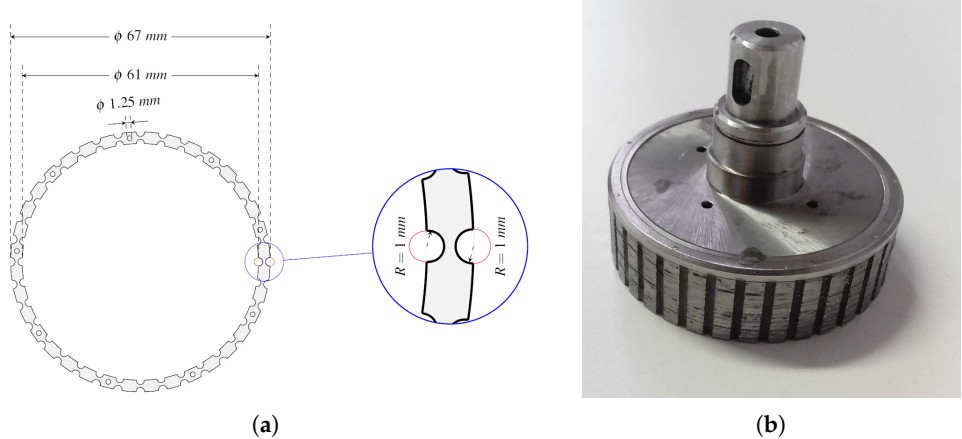

(**a**)          (**b**)

**Figure 4.** Primary shaft of the MRF clutch. Ferromagnetic layer dimensions (**a**) and prototype (**b**).

When the clutch is in the OFF-state (i.e., the PMs are far from the gap containing the fluid, see Figure 5a), the secondary shaft rotates with the camshaft and the shear-stress of the MRF is low: the speed of the primary shaft is low, and the vacuum pump is disengaged. In this configuration, the magnetic flux density inside the fluid is low, and the resulting low transmitted torque (up to 0.5 Nm) is mainly due to residual magnetization and parasitic effects. As experimentally assessed in [1], these latter effects are mainly due to the bearings and seals friction, approximately equal to 0.1 Nm at 1500 rpm.

As the distance between the PMs and the gap reduces, the magnetic induction in the MRF increases until the system reaches the ON-state (shown in Figure 5b). In this configuration, the clutch transmits a torque of about 5.5 Nm, high enough to engage the vacuum pump.

The specific design of the primary shaft, and in particular the presence of the double squirrel cage, allows the device to operate like an induction actuator when the two shafts rotate at different speeds (startup phase). In this phase, an additional electromagnetic torque generated from the interaction between the magnetic field of the rotating PMs

and the eddy currents flowing in the double squirrel cage can be produced. This torque, combined with the torque produced by the MRF, helps the clutch engagement. Once the clutch is engaged, the two shafts have the same rotational speed: the electromagnetic torque produced by the double squirrel cage vanishes, and the transmitted torque is only the MRF one.

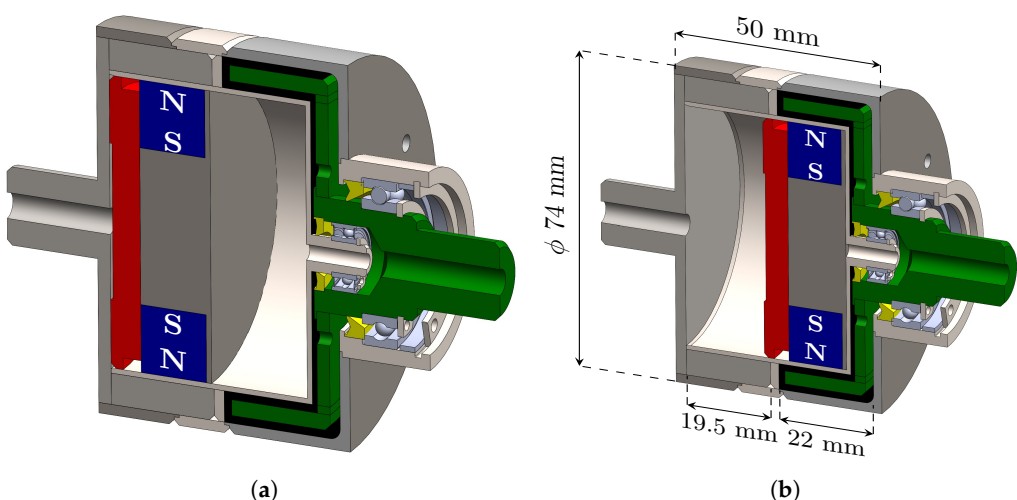

| (**a**) | (**b**) |
|---------|---------|

**Figure 5.** A 3D section view of the multi-gap MRF-based clutch. (**a**) OFF-state clutch; (**b**) ON-state clutch.

The device performance was investigated using a 3D dynamic Finite Element (FE) model based on the code EFFE. This software can consider the nonlinearity introduced by the ferromagnetic material and MRF, the presence of PMs, and the motion effect due to PMs' rotation around the device axis. The whole FE model of the structure contains about $2.5 \times 10^6$ elements and nodes.

The magnetic flux density inside the MR fluid was analyzed with some static simulations (i.e., fixing the relative speed to zero) in both OFF-state and ON-state conditions. Figure 6 shows the magnetic flux density distribution within the gap filled with the MRF. Despite the teeth and slots influencing the magnetic flux density's profile around the circumference, results indicate that this configuration effectively directs the magnetic flux lines across both the cylindrical parts of the MRF.

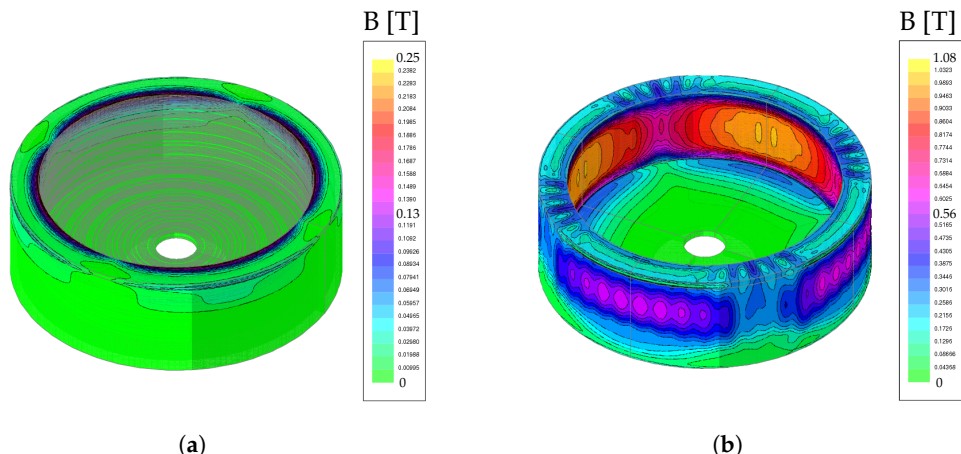

| (**a**) | (**b**) |
|---------|---------|

**Figure 6.** Distribution of the radial component of magnetic flux density within the MRF in the OFF-state (**a**) and ON-state (**b**) conditions.

Simulations under dynamic conditions were also performed to evaluate the contribution of the double squirrel cage to the performance of the device, smoothly increasing the speed of the primary shaft while blocking the secondary one. In this way, the relative

speed between the primary and secondary shafts increases to 1500 rpm in about 120 ms. Figure 7 shows the distribution of the eddy currents induced in the double squirrel cage of the primary shaft when the relative speed $\Delta\Omega = 1200$ rpm. Based on the simulation results, it was possible to confirm that the presence of the double squirrel cage, when the two shafts rotate at different speeds, generates an additional electromagnetic torque that facilitates the clutch engagement.

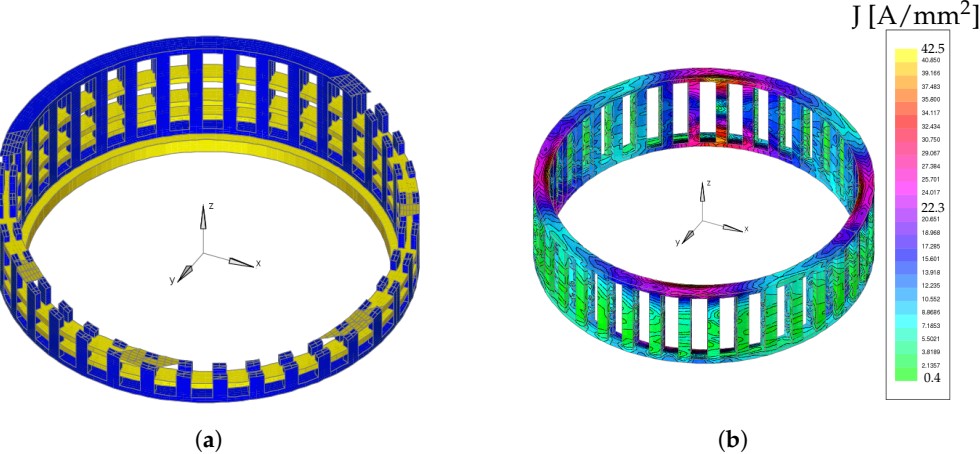

(**a**)    (**b**)

**Figure 7.** Three-dimensional (3D) FE model of the double squirrel cage (**a**) and distribution of the induced eddy current (**b**) in the primary shaft when the relative speed $\Delta\Omega = 1200$ rpm.

Figure 8 shows the final prototype of the clutch assembled. The fluid is injected into the chamber when the clutch is in the OFF-state or when the magnets are not inserted in the structure. In this way, the distance between the PMs and the fluid is sufficient to consider negligible the action of the PMs on the MRF so that the MRF behaves like a liquid.

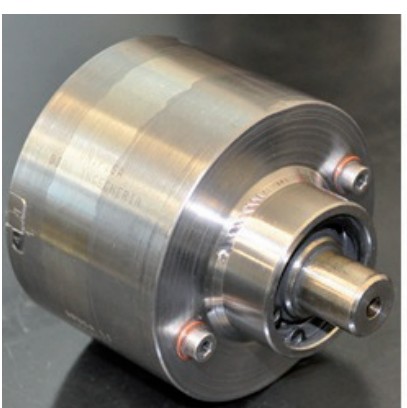 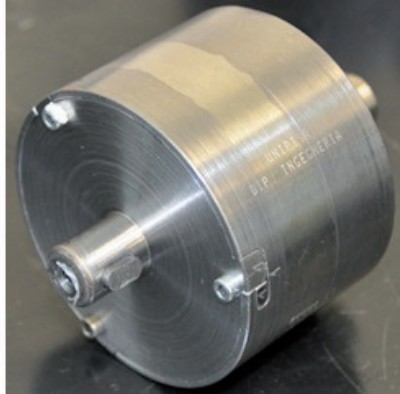

**Figure 8.** Prototype of the MRF clutch.

Measurements of the transmitted torque and the axial force as functions of the PMs displacement and relative speed $\Delta\Omega$ between the primary and secondary shaft were carried out to assess the operation of the device. Results are depicted in Figure 9. The blue curve in Figure 9a is the torque transmitted between the primary and secondary shafts with respect to the position of the PMs. The axial force, instead, is the attraction force between the PMs and the ferromagnetic materials that enclose them. This force either opposes or helps the PMs' movement during the engagement/disengagement phases, so it has to be evaluated carefully.

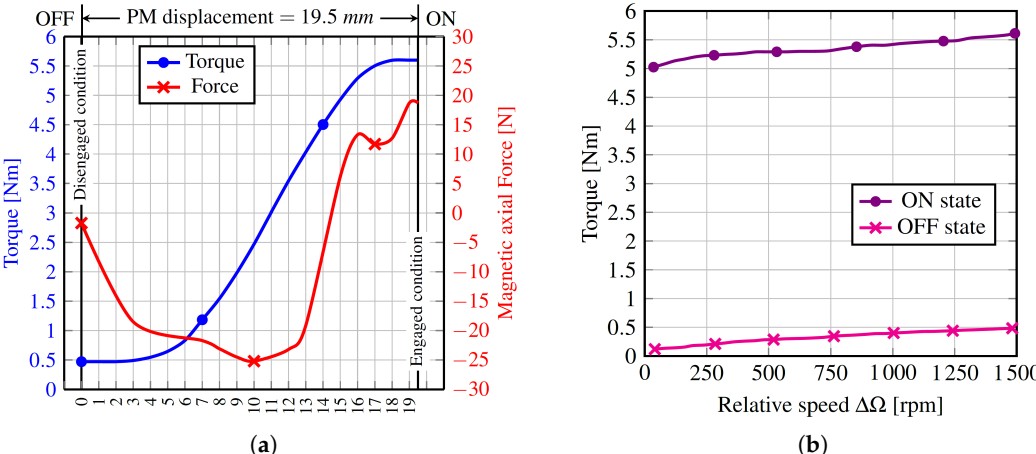

**Figure 9.** Torque and magnetic axial force as functions of the PMs displacement and relative speed. Experimental results. (**a**) Torque and axial force vs PMs displacement; (**b**) torque vs relative speed $\Delta\Omega$.

## 3. Experimental Tests

To experimentally assess the operation of the MRF-clutch in coupling/decoupling the vacuum pump from the camshaft, the device was tested, applying the standardized WLTC conditions. It is the standard driving cycle to evaluate the emission and fuel consumption performance of light-duty vehicles. It defines specific speed profiles that vehicles under test have to follow, which simulate the characteristics of a reference path. In particular, the WLTC for vehicle class 3 refers to the highest Power-to-Mass Ratio (PMR > 34 kW/ton) category and consists of four phases that mimic various driving scenarios, such as urban, non-urban, extra-urban, and high-speed driving conditions. In this context, we considered the 3.2 version applicable to vehicles with a maximum speed higher than 120 km/h (refer to Figure 10). During this experimental campaign, since some properties of the MRF could degrade with the temperature increase, the system was observed in its operation for one hour, considering two consecutive WLTCs to verify the effectiveness of the clutch in transmitting the required torque.

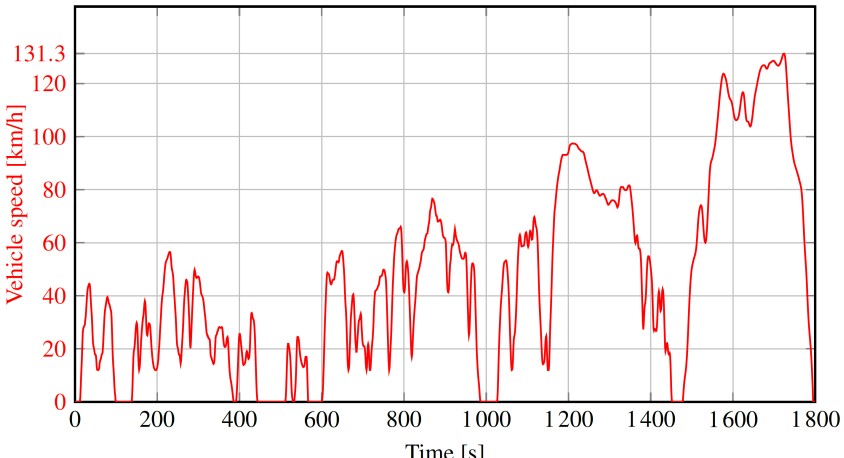

**Figure 10.** The standardized Worldwide harmonized Light-duty Test Cycle (WLTC).

A custom test bench was necessary to experimentally test the MRF-clutch and address the challenges related to coupling rotating components with the linear motor measuring torque and speed. A suitable support was 3D printed to transmit the axial force required during the clutch engagement/disengagement phases, and the decoupling of the rotating and linear motions was ensured with a bearing. However, this solution did not allow to directly connect the clutch's secondary shaft to a motor to simulate the camshaft. On the

other hand, transmitting motion through a belt and pulley directly to the clutch shaft would have resulted in transverse loads that would overload the internal clutch bearing. Therefore, the 3D-printed case with a housing for the belt shown in Figure 11 was designed to enclose the clutch and place radial ball bearings on both sides of the device. In this way, it was possible to transfer the forces resulting from the motion transmission to the supports without excessively stressing the clutch.

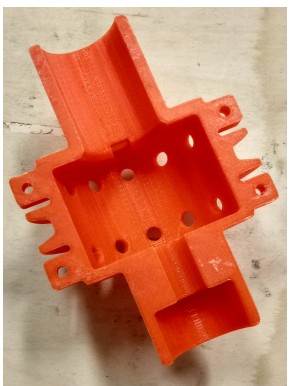 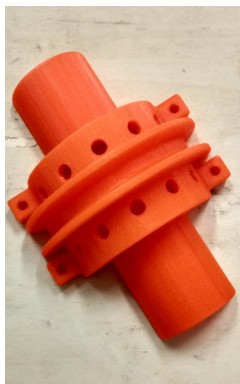

**Figure 11.** Parts of the 3D-printed case to enclose the clutch in the experimental setup.

The excitation system is axially moved with a linear stepper motor which replaces the pneumatic system used in the automotive application. This actuator is a LinMot linear motor with a maximum force of 585 N and a maximum speed of 1.6 m/s.

A speed-controlled brushless motor and a transmission belt simulate the camshaft's movement that imposes the rotation of the clutch's secondary shaft together with the PMs. The brushless motor has $V_n = 325$ V, $I_n = 3$ A and exerts a nominal torque $T_n = 4.5$ Nm.

The torque requested by the vacuum pump is not constant, as the depressor's torque varies as a function of the palette's angular position. This behavior is shown in Figure 12, where experimental data (@1500 rpm; 20 °C) were approximated with a sine wave, oscillating around a mean value of about 1.7 Nm. A torque-controlled DC motor, with a nominal power of 7.6 kW, and $V_n = 400$ V and $I_n = 22.5$ A, simulates the resistant torque applied by the depressor to the primary shaft of the clutch. The transmitted torque and rotational speed of the primary shaft are measured using a Kistler torque sensor with a range of 50 Nm positioned between the clutch and the DC motor. A DS Europe LT05 load cell with a 50 kg range acquires the axial force resulting from the interaction between the permanent magnets and the clutch ferromagnetic material components (including the MRF). Figure 13 shows the experimental setup used to test the clutch.

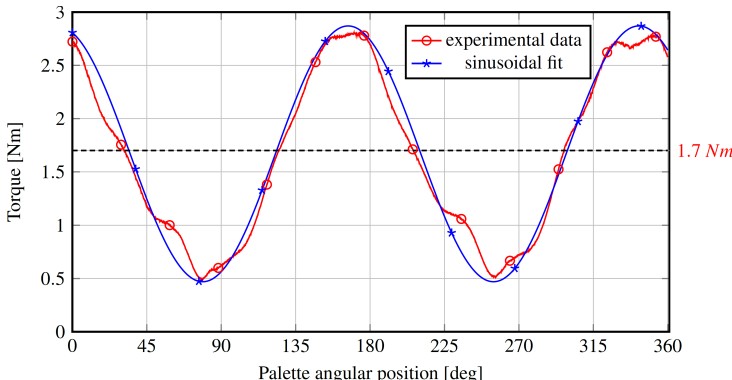

**Figure 12.** Torque requested by the vacuum pump as a function of the palette angular position (@ 1500 rpm; 20 °C).

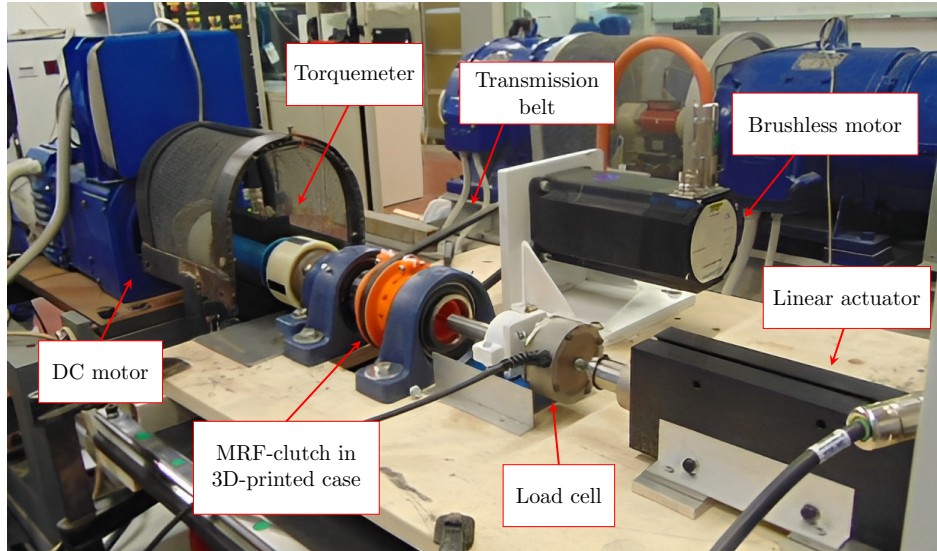

**Figure 13.** Test bench with actuation system for experimental validation.

Since the fluid is enclosed inside the clutch, it was impossible to directly measure the MRF temperature and observe how it influences the device's performance. To overcome this problem, a thin metallic layer with high thermal conductivity was fixed on the case that contains the device in direct contact with the external ferromagnetic case. Even though the temperature of the fluid certainly differs from the case one, after one hour of operation, we considered the structure at steady state thermal condition. An FLIR thermal camera was used to monitor the temperature increase during the experimental tests.

Before testing the MRF clutch under the WLTC conditions, preliminary measurements were performed to validate the experimental setup and the device's capabilities.

In particular, a first test was carried out to verify the maximum transmissible torque. Initially, the clutch was in the disengaged condition (OFF-state), with a null resistant torque applied by the DC motor and rotational speed of the primary shaft. The secondary shaft and the PMs excitation system were rotated at 1500 rpm with the brushless motor. Then, after about 35 s, the clutch was engaged (OFF- to ON-state), and once the primary shaft reached the same speed as the secondary one, the DC motor was controlled to increase (almost linearly) the resistant torque. At about 140 s, the torque reached the maximum value transmissible by the clutch (about 5.5 Nm), and the primary shaft abruptly decreased its speed. At this point, the DC motor reduced the resistant torque, reaching a value near 0.5 Nm. The rotational speed of the primary shaft grew again, reaching the speed of the secondary one until the system was turned off (at about 160 s). Figure 14 reports the measurements performed during this test. In particular, the transmitted torque is depicted in blue and the rotational speed of the primary shaft is in red.

The second test was performed to assess the system behavior during the engagement and disengagement phases. A constant resistant torque applied by the DC motor of about 4.5 Nm was considered. Again, the secondary shaft and the PMs rotated at 1500 rpm. As shown in Figure 15, the MRF clutch was initially in the OFF state (disengaged condition), and the speed of the primary shaft was zero. Once the clutch was engaged, after about 8 s, the primary shaft started to accelerate, reaching the same rotational speed as the secondary one in about 5 s. At a given time (approximately around 65 s), the clutch was disengaged, and the primary shaft speed decreased until it stopped.

The WLTC standard describes the test conditions in terms of vehicle speed but does not directly define the vacuum pump status. So, to test the device under the WLTC, it was necessary to determine the corresponding state condition of the clutch in every instant.

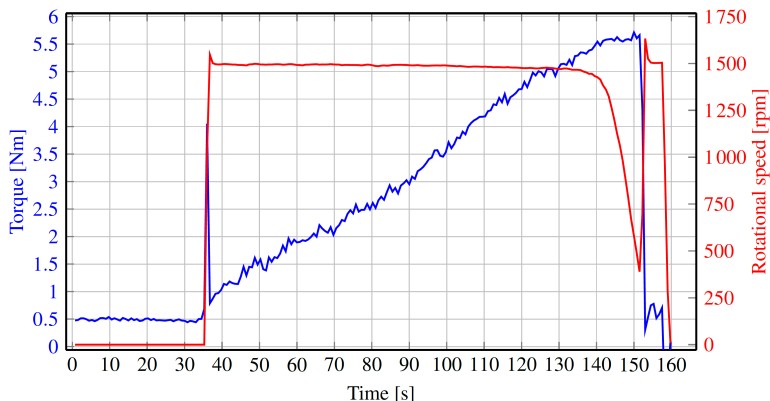

**Figure 14.** Maximum MRF clutch transmissible torque (in blue) and consequent rotational speed of the primary shaft (in red). Experimental measurements from the Kistler torque sensor.

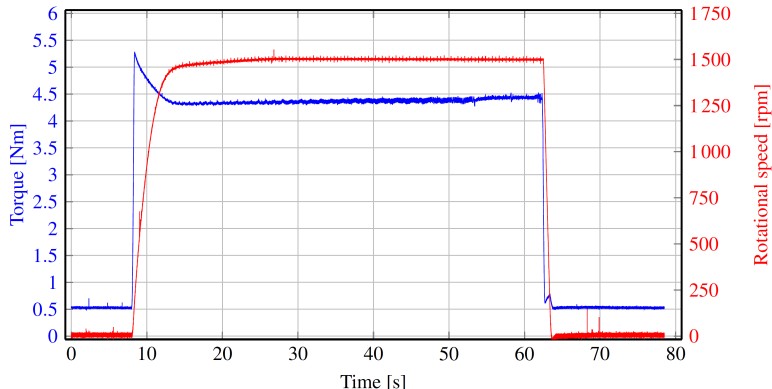

**Figure 15.** Engagement/disengagement phases with constant resistant torque of 4.5 Nm. The torque transmitted is shown in blue and the rotational speed of the primary shaft in red. Experimental measurements from the Kistler torque sensor.

In particular, the vacuum pump (and consequently the MRF-clutch) engagement phases were identified by finding the braking events during the WLTC. Every decrease in the WLTC's speed due to an instantaneous braking event causes the complete emptying of the booster chamber. Consequently, every time a braking event occurs, the vacuum pump has to be engaged for 20 s until its steady pressure value is reached. Figure 16 shows the MRF clutch status (ON—engaged/OFF—disengaged) overlapped with the WLTC.

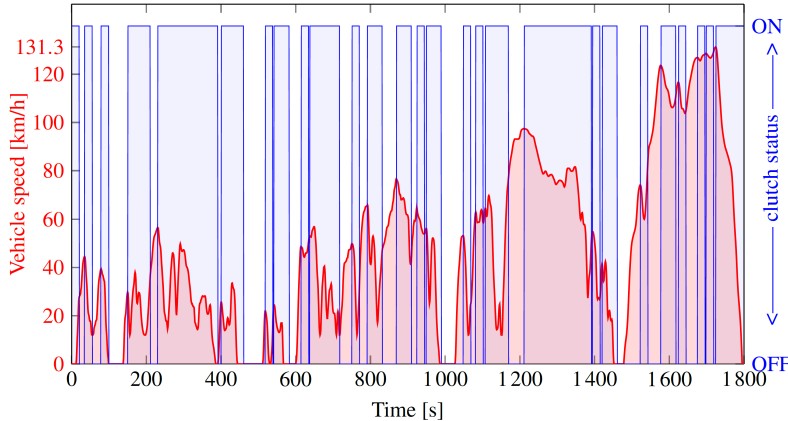

**Figure 16.** MRF clutch status as a function of the WLTC. The vehicle speed is shown in red and the clutch activation is highlighted in blue.

## 4. Results

The preliminary tests verified the capabilities of the MRF clutch and the correct design and assembly of the experimental test bench. Furthermore, the measured torque guarantees that in the engaged condition, there is no slip.

The system was tested for one hour, applying two consecutive standardized WLTCs with the resistant torque reported in Figure 12. The transmitted torque and the rotational speed of the primary shaft were continuously measured during the test, and the results are shown in Figure 17.

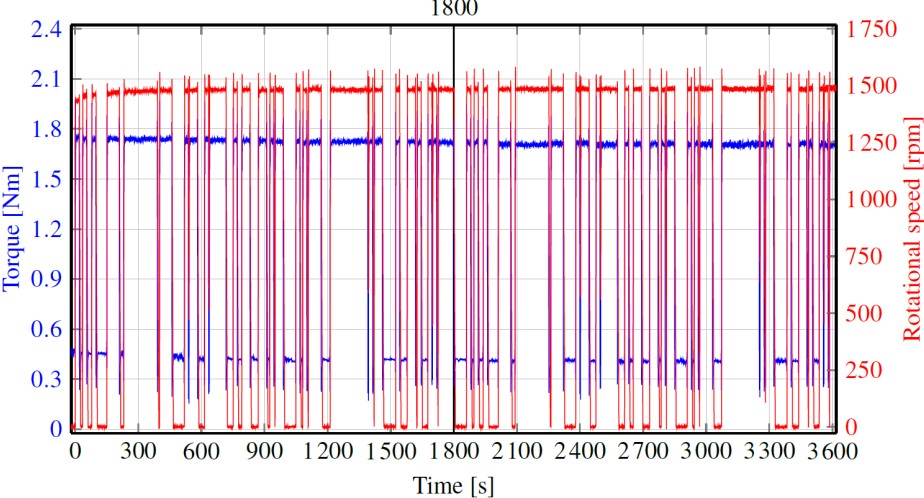

**Figure 17.** Behavior of the system under two consecutive WLTCs. MRF clutch transmitted torque (in blue) and rotational speed of the primary shaft (in red). Experimental measurements from the Kistler torque sensor.

Figure 18 shows the same quantities in the specific range of time $[600 \div 900]$ s. In particular, the profile of the WLTC vehicle speed is depicted in black, and crosses ($\times$) are placed in correspondence with braking events. As described in the Introduction section, the MRF clutch activation procedure expects that each braking event engages the clutch for 20 s. If a second braking event occurs during this period, the clutch does not change its status, but another time range of 20 s is added at the end of the previous interval unless an acceleration phase occurs before the ending of the preceding 20 s.

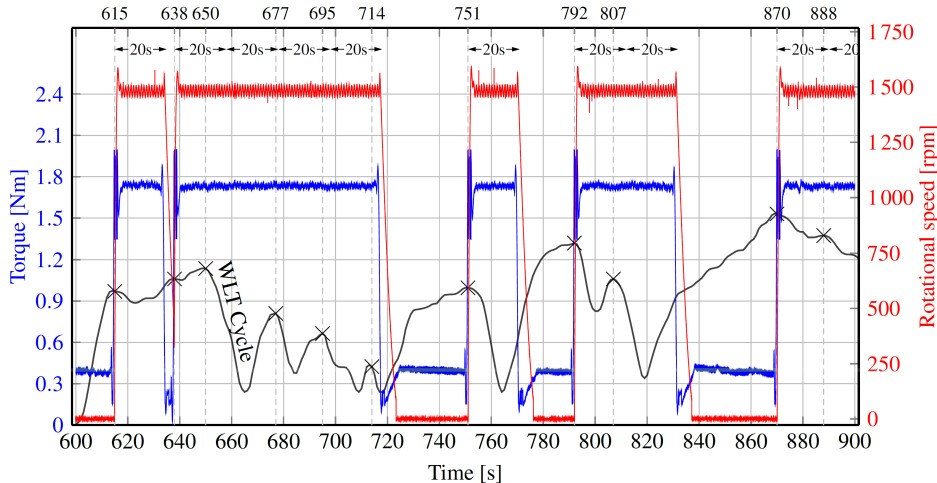

**Figure 18.** Behavior of torque and rotational speed of the clutch in the range $[600 \div 900]$ s of the WLTC. MRF clutch transmitted torque (in blue) and rotational speed of the primary shaft (in red). Experimental measurements from the Kistler torque sensor.

For example, referring to Figure 18, the braking event in $t = 638$ s activates the clutch for 20 s (until $t = 658$ s), but another braking event occurs before this time (in $t = 650$ s), making the device remain in the ON-state for an additional time of 20 s (until $t = 678$ s). The same happens with the following two braking events (in $t = 677$ s and $t = 695$ s), and the clutch stays engaged in the time interval $[678 \div 718]$ s. Different behavior can be observed focusing on the braking event that takes place in $t = 714$ s. Since the device is already in the ON-state at this instant, 20 more seconds would be added (until $t = 738$ s), but, in correspondence with the ending of the preceding 20 s (i.e., in $t = 718$ s), an acceleration phase of the WLTC occurs. For this reason, in $t = 718$ s, the MRF clutch disengages.

During the test, the temperature was monitored to detect variations in the MRF operating conditions using an FLIR thermal camera. In the beginning, the temperature of the external case of the clutch was about 27 °C, and after one hour of operation, it increased to 75 °C. Overall, after two consecutive WLTCs, the temperature of the clutch (and consequently, the working temperature of the MRF) increased by about 48 °C.

The two thermal conditions of the system are shown in Figure 19.

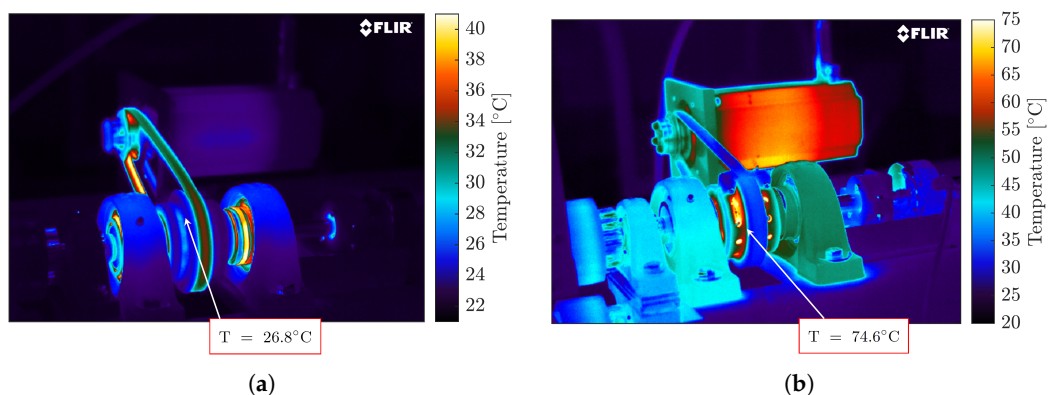

**Figure 19.** Thermal camera images of the MRF clutch and the neighboring parts of the test bench. The clutch temperature is pointed out in the beginning of the test (**a**) and after one hour of testing (**b**).

## 5. Discussion

The measurements obtained show that the performance of the system is only marginally affected by its operation under two consecutive WLTCs: the transmitted torque after one-hour testing has been reduced by a mean value of about 2% in the engaged condition (ON-state) and about 10% in the disengaged one (OFF-state). The torque reduction in the OFF-state condition is mainly due to the temperature increase. Temperature affects the viscosity parameter of the MRF, whose contribution is predominant when the fluid is not excited by the magnetic field. On the contrary, during the engaged condition, the contribution of the magnetic field-induced torque is much higher than that of the viscosity torque, and the temperature increase has little effect on the transmitted torque.

## 6. Conclusions

In this paper, the performance of an MRF clutch used to engage/disengage the vacuum pump connected to the power-brake unit was analyzed. A custom test bench was designed, and the device was experimentally tested under two consecutive standardized WLTCs, operating continuously for one hour. As the temperature increase affects the performance of the MRF, an FLIR thermal camera was used to monitor the system during the test.

Following the specification of the envisaged application, the clutch was efficient in transmitting a given torque when engaged (5.5 Nm) and having a little dissipation if disengaged (0.5 Nm). The value of torque after one hour of operation was the same as that transmitted in the beginning in both the working conditions of the clutch (OFF-state and ON-state), highlighting the marginal effect of the temperature increase. The obtained

results proved the effectiveness of the developed MRF clutch for the selected application (i.e., the vacuum pump in a diesel engine) in the automotive framework, in compliance with the constraints of a small volume and fail-safe operation.

**7. Patents**

1. **Mechanical combustion engine driven fluid pump**, (2011), (EP20110425176 20110704; WO2012EP51006 20120124; EP2543903 (A1) EP2543903 (B1) JP2014521022 (A) JP5808486 (B2) WO2013004401 (A1) CN103635710 (A) CN103635710 (B)).
2. **Mechanical combustion engine driven fluid pump, a combined Permanent Magnet Magneto-Rheological/Eddy Currents Clutch**, (2012/2015), (WO2014029445 (A1) JP2015529312 (A) EP2888496 (A1) EP2888496 (B1) CN104884834 (A)).
3. **Mechanical combustion engine driven fluid pump, a Fail-safe Magneto-Rheological Multidisk Clutch Activated by Permanent Magnets**, (2012/2015), (WO2012EP66465 20120823 WO2014029446 (A1) JP2015527546 (A) EP2888497 (A1) CN104603490 (A)).

**Author Contributions:** Conceptualization, R.R., C.S. and A.M.; methodology, A.M. and R.R.; software, L.S. and C.S.; validation, R.R., C.S. and L.S.; formal analysis, A.M. and C.S.; investigation, M.F.-M. and N.G.; resources, R.R.; data curation, C.S. and M.F.-M.; writing—original draft preparation, C.S. and R.R.; writing—review and editing, N.G. and L.S.; visualization, C.S.; supervision, R.R., A.M. and L.S.; project administration, R.R.; funding acquisition, R.R. All authors have read and agreed to the published version of the manuscript.

**Funding:** This research was funded: by National Recovery and Resilience Plan (NRRP), CN1, Centro Nazionale di Ricerca in High Performance Computing, Big Data e Quantum Computing, spoke 6: multiscale modelling & engineering applications; and also by Pierburg Pump Technology Italy, S.p.A., within a framework of a project supported by Regione Toscana, P.O.R., C.R.e.O., F.E.S.R. 2007–2013.

**Institutional Review Board Statement:** Not applicable.

**Conflicts of Interest:** The authors declare no conflict of interest.

**Abbreviations**

The following abbreviations are used in this manuscript:

| | |
|---|---|
| FE | Finite Element |
| MRF | MagnetoRheological Fluid |
| PM | Permanent Magnet |
| WLTC | Worldwide harmonized Light-duty Test Cycle |

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
