# Peer review of "Experimental Validation of a Permanent Magnets Magnetorheological Device under a Standardized Worldwide Harmonized Light-Duty Test Cycle"

_actuators, doi:10.3390/act12100375_

Round 1

Reviewer 1 Report

The paper describes design, simulation, experimental realization and standardized testing of a magnetorheological clutch for selective activation and deactivation of a vacuum pump for power-braking systems in automobiles. To the best of my knowledge, the work is novel and original. The title of the paper is appropriate. The abstract summarizes well the motivation, the approach and the most important test results. In the introduction, the work is motivated by the necessity for energy-saving automobile components, and the relevance of the work is laid out in great detail. The current approach is nicely put into context with prior research in that field. The properties and advantages of magnetorheological fluids are comprehensively explained and nicely compared to their electrical counterpart electrorheological fluids. The design details of the multi-gap clutch are precisely described and nicely illustrated by technical drawings and photographs. The clutch is realized by a magnetizable cylinder on the primary shaft, rotating in the magnetorheological fluid, into which a permanent magnet rotor connected to the secondary shaft can be inserted or removed by air-pressure activation. A simulation of the magnetic field in the magnetorheological fluid illustrates the principle of operation. The torque between both rotors was measured as a function of magnet displacement and of relative speed. The test bench constructed for performing the standardized test cycle (WLTC) is explained, supported by a photograph of the setup. Torque and temperature of the components are continuously monitored. The results of the WLTC tests confirmed reliable operation of the clutch under the prescribed realistic test conditions, proving its applicability for power-braking systems in automobiles.

I only found one (possible) typo in line 221, I think “if” should be replaced by “though”.

I think it is uncommon to have just one sub-heading in a chapter, usually if you go to the next level of sub-headings, there should be at least two sub-chapters. Therefore, the authors might consider either removing sub-heading “1.1. Magnetorheological Fluids” or, better, adding a new sub-heading, e.g. between lines 19 and 20, which could for instance be called “1.1. Motivation”.

Reviewer 2 Report

In this paper, the MRF clutch was tested using a standardized Worldwide harmonized Light-duty Test Cycle (WLTC) performed on a test bench. The system was observed for one hour during its operation, considering two consecutive WLTCs. The results were analyzed to verify the effects of a standardized procedure on the device’s performance. The paper is interesting and contains some new ideas. I recommend this paper to be published after the following revisions.

Point 1: In the introduction section, there is too much review space on MR fluids, and it is recommended to delete some of the content. Additionally, more recent articles on the structure of MRF clutches should be reviewed, which will be helpful to highlight the novelty of the structures proposed in this article.

Point 2: In section 2.1, the working principle of the proposed MRF clutch should be described in detail. Meanwhile, please indicate the magnetic force line direction of the MRF clutch in Figure 2.

Point 3: In Figure 11(a), the authors analyzed the variation of transmitted torque and axial force with PMS displacement. To avoid confusion between torque and force, a mathematical relationship should be given.

Point 4: In the early stage of experimental testing, how did the authors inject MRF into the MRF clutch? According to the reviewer's understanding, MRF will transform into a quasi-solid state within milliseconds under the action of a magnet.

Point 5: How to measure the frictional damping torque between the seal and rotating shaft? This is not negligible.
